# GUI-R1: A GENERALIST R1-STYLE VISION-LANGUAGE ACTION MODEL FOR GUI AGENTS

## ABSTRACT

Existing efforts in building graphical user interface (GUI) agents largely rely on the training paradigm of supervised fine-tuning (SFT) on large vision-language models (LVLMs). However, this approach not only demands extensive amounts of training data but also struggles to effectively understand GUI screenshots and generalize to unseen interfaces. The issue significantly limits its application in real-world scenarios, especially for high-level tasks. Inspired by reinforcement fine-tuning (RFT) in large reasoning models (*e.g.*, DeepSeek-R1), which efficiently enhances the problem-solving capabilities of large language models in real-world settings, we propose GUI-R1, the first reinforcement learning framework designed to enhance the GUI capabilities of LVLMs in high-level real-world task scenarios, through unified action space rule modeling. By leveraging a small amount of carefully curated high-quality data across multiple platforms (including Windows, Linux, MacOS, Android, and Web) and improved policy optimization algorithms to update the model, GUI-R1 achieves superior performance using only 0.02% of the data (3K vs. 13M) compared to previous state-of-the-art methods like OS-Atlas across eight benchmarks spanning three different platforms (mobile, desktop, and web). These results demonstrate the immense potential of reinforcement learning based on unified action space rule modeling in improving the execution capabilities of LVLMs for real-world GUI agent tasks. We will fully open-source GUI-R1 to benefit the research field.

## 1 INTRODUCTION

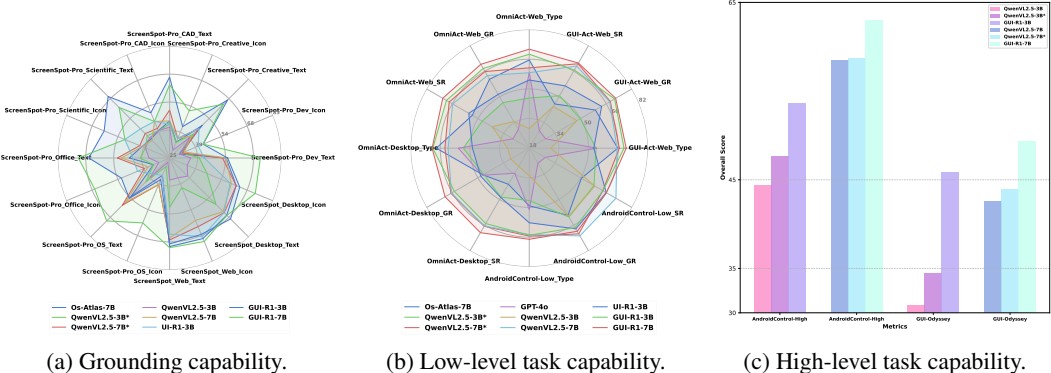

|                             |                             |                             |
| :-------------------------: | :-------------------------: | :-------------------------: |
| (a) Grounding capability.   | (b) Low-level task capability. | (c) High-level task capability. |

Figure 1: GUI-R1 achieves the best performance on eight evaluation datasets covering various platforms and task granularities, demonstrating the promising potential of RFT in GUI agent tasks.

Recent studies (Wu et al., 2024b; Qin et al., 2025; Cheng et al., 2024; Wang et al., 2024b) have explored the use of large vision-language models (LVLMs) (Bai et al., 2025) to develop graphical user interface (GUI) agents capable of performing high-level complex tasks. These agents analyze the screen as a self-contained source of information for decision-making, without relying on environment-based textual descriptions such as HTML or accessibility trees. This approach offers

greater flexibility in agent decision-making. However, previous works have predominantly relied on the training paradigm of supervised fine-tuning (SFT), which not only requires large amounts of high-quality training data but also struggles to effectively comprehend GUI screenshots and generalize to unseen interfaces. These limitations have significantly hindered the real-world applicability of these works, particularly for high-level GUI tasks that lack explicit step-by-step instructions.

Rule-based reinforcement fine-tuning has recently emerged as an efficient and scalable alternative to SFT, requiring only a small number of examples to fine-tune models effectively while demonstrating strong performance and generalization capabilities in domain-specific tasks. RFT has been increasingly adopted for developing various LVLMs (Liu et al., 2025c; Huang et al., 2025; Chen et al., 2025b; Shen et al., 2025; Lu et al., 2025a; Chen et al., 2025a; Pan et al., 2025; Deng et al., 2025; Peng et al., 2025; Zhang et al., 2025c). Inspired by these advancements, this study extends the rule-based reinforcement learning (RL) paradigm to the domain of GUI agents, which focuses on GUI action prediction tasks within a unified action space driven by high-level instructions. Specifically, LVLMs generate multiple responses (trajectories) for each input, containing both reasoning traces and final answers. These responses are evaluated using a unified action space reward function designed in this work, with Gaussian-based click verification as one component, and the model is updated through policy optimization (Guo et al., 2025).

To further enhance the effectiveness of RL in GUI grounding, we introduce GUI-GRPO, which incorporates two key improvements: (1) a maximum entropy constraint to encourage exploration, addressing the tendency of standard GRPO (Guo et al., 2025) to collapse around UI element centers when using continuous Gaussian click verification rewards, and (2) sentence-level training granularity to mitigate length bias introduced by token-level optimization, ensuring that reasoning traces are properly leveraged while maintaining high-quality action predictions. This combination enables more stable policy learning, better reasoning capabilities, and improved generalization to out-of-distribution (OOD) GUI scenarios. By modeling a unified action space, we efficiently curate high-quality data spanning multiple platforms, including Windows, Linux, MacOS, Android, and Web, while avoiding action prediction conflicts across different platforms.

As demonstrated in Figure 1, the proposed framework (GUI-R1) achieves superior performance using only 0.02% of the data (3K vs. 13M) compared to previous state-of-the-art methods like OS-Atlas (Wu et al., 2024b) across eight benchmarks covering three different platforms (mobile, desktop, and web) and three levels of task granularity (low-level grounding, low-level tasks, and high-level tasks). Before delving into details, we clearly emphasize our contribution as follows.

- We propose GUI-R1, the first framework that utilizes rule-based reinforcement fine-tuning to enhance the reasoning capabilities of LVLMs in high-level GUI action prediction tasks.

- We develop a rule-based unified action space featuring a Gaussian reward function with an enhanced GUI-GRPO algorithm, which efficiently validates GUI task responses across different platforms and task granularities. This ensures reliable and efficient data selection and model training.

- Leveraging the rule-based unified action space reward function, we construct GUI-R1-3K, which is a high-quality fine-tuning dataset with diversity and complexity. This dataset significantly improves both training efficiency and model performance.

- We conduct a comprehensive evaluation of GUI agents, covering three distinct platforms (desktop, mobile, and web) and three levels of task granularity (low-level grounding, low-level tasks, and high-level tasks) across eight benchmarks. Experimental results demonstrate that our GUI-R1 is leading in multiple realistic cases. This creates a strong baseline of GUI agents for future research. We will fully open-source GUI-R1 to benefit the research community and accelerate the advancement of GUI agent development.

## 2 GUI-R1 FRAMEWORK

GUI-R1 is based on a reinforcement learning training paradigm designed to enhance the ability of GUI agents to complete sophisticated instructional tasks. As shown in Figure 2, unlike low-level tasks, high-level GUI tasks lack explicit and fine-grained instructions, which require action predictions based on high-level task objectives and execution history. This imposes greater demands on the model's contextual learning and understanding capabilities.

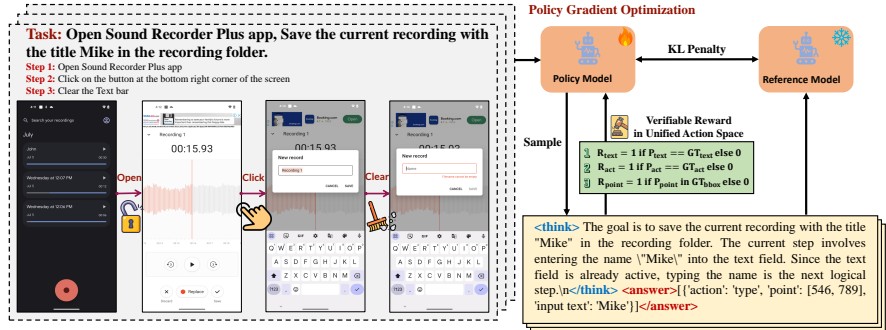

Figure 2: **Overview of the GUI-R1 Framework.** Given the high-level instruction, action history, and visual image inputs, the policy model generates multiple responses containing reasoning steps. Then the verifiable rewards, such as action type reward, click point reward, and input text reward, are used with the policy gradient optimization algorithm to update the policy model.

## 2.1 PRELIMINARIES

We define the goal of GUI agents in high-level instructional tasks as understanding and executing low-level instructions to complete the high-level task $q$, based on the current interface image $x$, and the execution history $h$. Formally, given the input $q$, $x$, and $h$, the model generates a set of candidate responses $O = \{o_1, o_2, \ldots, o_N\}$, where each response contains attributes of the predicted low-level action $o^{\text{act}}$, input text $o^{\text{text}}$, and input point $o^{\text{point}}$. Each response is evaluated using a unified action space reward function $R$ to compute its reward $\{R(q, o_1), R(q, o_2), \ldots, R(q, o_N)\}$. Reinforcement learning with verifiable rewards (RLVR) methods, such as the widely adopted GRPO (Guo et al., 2025) algorithms, typically optimize the following objective:

$$\mathcal{J}(\theta) = \mathbb{E}_{q \sim P(Q), \, o \sim \pi_\theta(\cdot|q)} \left[ R(q, o) \right], \tag{1}$$

where $q$ denotes a query sampled from the distribution $P(Q)$, $o$ is an output generated by the policy $\pi_\theta$ which is parameterized by $\theta$. The GRPO objective function is formulated as:

$$\mathcal{J}_{\text{GRPO}}(\theta) = \mathbb{E}_{q \sim P(q), \{o_i\}_{i=1}^N \sim \pi_{\theta_{\text{old}}}(\cdot|q)} \left[ \frac{1}{N} \sum_{i=1}^{N} \frac{1}{|o_i|} \sum_{t=1}^{|o_i|} L_{i,t} \right], \tag{2}$$

The loss term $L_{i,t}$ is defined as:

$$L_{i,t} = \min\left[ r_{i,t} A_{i,t}, \text{clip}(r_{i,t}, 1 - \varepsilon, 1 + \varepsilon) A_{i,t} \right] - \beta D_{\text{KL}}[\pi_\theta \| \pi_{\text{ref}}], \tag{3}$$

where $\pi_{\text{ref}}$ represents the reference policy, and $\text{clip}(\cdot, 1 - \varepsilon, 1 + \varepsilon)$ applies clipping to stabilize training. The probability ratio $r_{i,t}$ and advantage function $A_{i,t}$ are calculated as:

$$r_{i,t} = \frac{\pi_\theta(o_{i,t}|q, o_{i,<t})}{\pi_{\theta_{\text{ref}}}(o_{i,t}|q, o_{i,<t})}, A_{i,t} = \frac{R(q, o_i) - \mu_R}{\sigma_R}, \tag{4}$$

where $\mu_R = \text{mean}(\{R(q, o_i)\}_{i=1}^N)$ and $\sigma_R = \text{std}(\{R(q, o_i)\}_{i=1}^N)$ are the mean and standard deviation of rewards across all generated outputs. The KL divergence term is calculated as:

$$D_{\text{KL}}[\pi_\theta \| \pi_{\text{ref}}] = \frac{\pi_{\text{ref}}(o_{i,t}|q, o_{i,<t})}{\pi_\theta(o_{i,t}|q, o_{i,<t})} - \log \frac{\pi_{\text{ref}}(o_{i,t}|q, o_{i,<t})}{\pi_\theta(o_{i,t}|q, o_{i,<t})} - 1 \tag{5}$$

Note that the KL divergence term can be omitted by setting $\beta = 0$ if regularization is not required.

## 2.2 VERIFIABLE REWARDS IN UNIFIED ACTION SPACE

We adopt a unified action space modeling strategy, which extracts action space categories across different platforms and integrates them into a unified action space. This ensures that all high-level instructions can be decomposed into a sequence of atomic actions, resolving action space conflicts in multi-platform data joint training. Based on the unified action space, we design verifiable reward

functions to evaluate the accuracy of predicted actions to guide reinforcement learning. Notably, instead of utilizing the discrete norm reward function as shown in Figure 2, we design a continuous Gaussian reward that can provide more precise reward. We detail these verifiable rewards below.

**Format reward.** Following previous work (Meng et al., 2025; Guo et al., 2025; Huang et al., 2025), we introduce format rewards during training to evaluate whether the generated output adheres to the expected structural format, including both syntactic and semantic validity. Specifically, format rewards guide the model to generate reasoning processes and final answers in a structured format, which play a critical role in self-learning and iterative improvement during reinforcement fine-tuning. The format reward templates used in training and inference are as follows, where '<think>' represents the reasoning process and '<answer>' represents the final answer.

---

**Unified Action Space Prompt for Task Training and Inference**

You are GUI-R1, a reasoning GUI Agent Assistant. In this UI screenshot $< image >$, I want you to continue executing the command $task$, with the action history being $history$. Please provide the action to perform (enumerate from [complete, close/delete, press_home, click, press_back, type, select, scroll, enter]), the point where the cursor is moved to (integer) if a click is performed, and any input text required to complete the action.

Output the thinking process in $<think> </think>$ tags, and the final answer in $<answer> </answer>$ tags as follows: $<think>$ ... $</think>$ $<answer>$['action': enum[complete, close/delete, press_home, click, press_back, type, select, scroll, enter], 'point': [x, y], 'input_text': 'no input text [default]']$</answer>$.

---

**Unified Action Space Prompt for Grounding Training and Inference**

You are GUI-R1, a reasoning GUI Agent Assistant. In this UI screenshot $< image >$, I want you to continue executing the command $task$, with the action history being $history$. Please provide the action to perform (enumerate from [click]), the point where the cursor is moved to (integer) if a click is performed, and any input text required to complete the action.

Output the thinking process in $<think> </think>$ tags, and the final answer in $<answer> </answer>$ tags as follows: $<think>$ ... $</think>$ $<answer>$['action': enum[click], 'point': [x, y], 'input_text': 'no input text [default]']$</answer>$.

---

**Accuracy rewards.** For the model's predicted output $o = \{o^{\text{act}}, o^{\text{text}}, o^{\text{point}}\}$, which consists of three components: $o^{\text{act}}$ (action type, *e.g.*, click, scroll), $o^{\text{point}}$ (click point position), and $o^{\text{text}}$ (input text), we define the accuracy reward $R_{\text{acc}}$ as a combination of action type reward $R_{\text{act}}$, click point reward $R_{\text{point}}$, and input text reward $R_{\text{text}}$, *i.e.*, $R_{\text{acc}} = R_{\text{act}} + R_{\text{point}} + R_{\text{text}}$. This design provides reliable correctness rewards for all actions.

**Action type reward.** The action type reward $R_{\text{act}}$ is calculated by comparing the predicted action type $o^{\text{act}}$ with the ground truth action type $gt^{\text{act}}$. If $o^{\text{act}} == gt^{\text{act}}$, the reward is 1; otherwise, it is 0. This simple yet effective evaluation mechanism guides action type prediction.

**Gaussian click point reward.** The click point reward $R_{\text{point}}$ is calculated by comparing the L2 norm distance $d$ between predicted click point $o^{\text{point}} = [\text{x}, \text{y}]$ and the center of ground truth bounding box $gt^{\text{bbox}} = [\text{x}_1, \text{y}_1, \text{x}_2, \text{y}_2]$. The calculation formula is as follows:

$$R_{\text{point}} = \begin{cases} \exp(-d_{\text{point}}^2) & \text{if } o^{\text{point}} \in gt^{\text{bbox}}, \\ 0 & \text{otherwise.} \end{cases}$$

**Gaussian input text reward** The input text reward $R_{\text{text}}$ is calculated by comparing the predicted input text $o^{\text{text}}$ with the ground truth text parameter $gt^{\text{text}}$ using the semantic $F_1$ score. The calculation formula is as follows:

$$R_{\text{text}} = \begin{cases} \exp(-(1 - F_1(o^{\text{text}}, gt^{\text{text}}))) & \text{if } F_1(o^{\text{text}}, gt^{\text{text}}) > 0.5, \\ 0 & \text{otherwise.} \end{cases}$$

**Response reward.** The final response reward is composed of format rewards and accuracy rewards dynamically, defined as: $R_o = \alpha R_{\text{f}} + \beta R_{\text{acc}}$, where $R_{\text{f}}$ represents the format reward, $R_{\text{acc}}$ represents the accuracy reward, and $\alpha$ and $\beta$ are weighting parameters respectively.

## 2.3 GUI-GRPO

To efficiently enhance the reinforcement learning (RL) effectiveness for GUI-Grounding, we design GUI-GRPO with two key improvements to enhance the performance of GRPO: 1) Adding maximum entropy constraint terms to encourage exploration. Due to the adoption of continuous Gaussian click verification reward functions, GRPO during training tends to explore only around the center points of UI elements to obtain higher rewards, easily falling into entropy collapse dilemmas that limit further performance improvements. We incorporate maximum entropy constraints into the optimization to encourage model exploration during learning, enabling more effective learning. 2) Changing the training granularity from token-level to sentence-level. We find that in reinforcement learning training for improving GUI grounding capabilities, token-level GRPO training easily introduces length bias. Specifically, most training data consists of click action grounding reward verification, where the thinking process of these samples has minimal impact on final coordinate prediction accuracy. Moreover, overly long response samples tend to have lower quality, and token-level optimization approaches introduce additional bias, resulting in suboptimal effects.

We consider augmenting the standard objective with an entropy term, leading to a maximum entropy RL formulation:

$$\mathcal{J}_{\text{GUI-GRPO}}(\theta) = \mathbb{E}_{q \sim P(Q),\ o \sim \pi_\theta(\cdot|q)} \big[ R(q, o) + \alpha \mathcal{H}\left(\pi_\theta(\cdot|q)\right) \big], \tag{6}$$

where $\mathcal{H}(\pi_\theta(\cdot|q)) = \pi_\theta(\cdot|q) \log \pi_\theta(\cdot|q)$ denotes the entropy of the policy and encourages exploration, while the coefficient $\alpha \geq 0$ balances the trade-off between reward maximization and entropy regularization. When $\alpha = 0$, it reduces to Eq. (1).

Given the maximum entropy RL objective above, we now follow the standard policy gradient framework to derive the policy gradient of the maximum entropy objective $\mathcal{J}_{\text{GUI-GRPO}}(\theta)$:

$$\nabla_\theta \mathcal{J}_{\text{GUI-GRPO}}(\theta) = \nabla_\theta \mathbb{E}_{q \sim P(Q),\ o \sim \pi_\theta(\cdot|q)} \big[ R(q, o) + \alpha \mathcal{H}\left(\pi_\theta(\cdot|q)\right) \big] \tag{7}$$

$$= \mathbb{E}_{q \sim P(Q), o \sim \pi_\theta(\cdot|q)} \Big[ \nabla_\theta \log \pi_\theta(o|q) \big( R(q, o) - \alpha \log \pi_\theta(o|q) \big) \Big]. \tag{8}$$

We now define the objective function for GUI-GRPO:

$$\mathcal{J}_{\text{GUI-GRPO}}(\theta) = \mathbb{E}_{q \sim P(Q), \{o_i\}_{i=1}^N \sim \pi_{\theta_{\text{old}}}(\cdot|q)}$$

$$\frac{1}{N} \sum_{i=1}^N \frac{1}{|o_i|} \sum_{t=1}^{|o_i|} \left\{ \min \left[ \frac{\pi_\theta(o_{i,t}|q, o_{i,<t})}{\pi_{\theta_{\text{old}}}(o_{i,t}|q, o_{i,<t})} \bar{A}_{i,t}, \text{clip}\left( \frac{\pi_\theta(o_{i,t}|q, o_{i,<t})}{\pi_{\theta_{\text{old}}}(o_{i,t}|q, o_{i,<t})}, 1 - \varepsilon, 1 + \varepsilon \right) \bar{A}_{i,t} \right] \right\}, \tag{9}$$

where $\pi_{\theta_{\text{old}}}$ denotes the old policy. We afterward calculate $\bar{A}_{i,t}$ at *sequence-level*:

$$\bar{A}_{i,t} = A_{i,t} + \alpha_q \cdot \psi_i(\pi_\theta), \tag{10}$$

where

$$\psi_i(\pi_\theta) = -\log \pi_\theta^{\text{detached}} \text{mean}_{\text{batch}}(\{-\log \pi_\theta^{\text{detached}}, \dots\}) \tag{11}$$

and

$$\log \pi_\theta^{\text{detached}}(o_i|q) = \frac{1}{|o_i|} \sum_{t=1}^{|o_i|} \log \pi_\theta^{\text{detached}}(o_{i,t}|q, o_{i,<t}), \tag{12}$$

where $\alpha_q$ is a coefficient weight updated by an exponential moving average (EMA) dynamically.

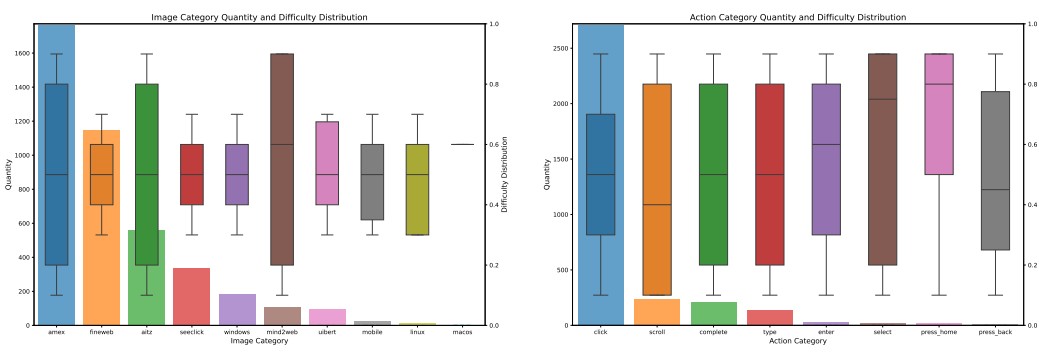

(a) Image category quantity and difficulty distribution.  (b) Action category quantity and difficulty distribution.

Figure 3: Illustrations of image and action category quantity and difficulty distributions in the dataset GUI-R1-3K.

## 3 EXPERIMENTS

### 3.1 TRAINING DATA CURATION

**Data collection.** We collect data related to GUI tasks from multiple open-source datasets, including FineWeb (Penedo et al., 2024), UIBert (Bai et al., 2021), AMEX (Chai et al., 2024), RICOSCA (Li et al., 2020), as well as portions of pretraining data from Seeclick (Cheng et al., 2024) and OS-Otlas (Wu et al., 2024b). This leads to ∼14M examples of grounding and low-level task data. Additionally, we collect ∼30K high-level GUI data points from OS-Otlas instruction datasets. In total, we gather ∼14M examples spanning multiple platforms (including Windows, Linux, MacOS, Android, and Web) and various task granularities (grounding, low-level, and high-level).

**Data filtering.** To filter out low-quality data for efficient RFT, we use the Qwen2.5VL-7B (Bai et al., 2025) model to generate 10 responses for each example and evaluate them using a rule-based reward function designed for unified action space modeling. We remove the problems with an estimated accuracy of 0 or 1 to ensure a stable training process, resulting in 140K low-level data and 1.5K high-level data. Since the quantity of low-level data far exceeds that of high-level data, we randomly sample 1.5K low-level data and combine it with all high-level data to create a balanced dataset of 3K high-quality training examples, named GUI-R1-3K. The distribution of image categories, action types, and corresponding difficulty levels is demonstrated in Figure 3.

### 3.2 IMPLEMENTATION DETAILS

**Training and inference details.** For supervised fune-tuning (SFT), we use the QwenVL2.5-3B/7B (Bai et al., 2025) model as the base model for experiments and employ the LLaMA Factory (Zheng et al., 2024) framework for one epoch of training to avoid overfitting. For RFT, we use the EasyR1 (Zheng et al., 2025) framework for training over nine epochs. During inference, to ensure fairness, we apply a unified and simple prompt across all comparison methods, and conduct experiments under zero-shot prompt configurations. All experiments are conducted using 8×NVIDIA A100-80G GPUs.

**Evaluation benchmarks.** We evaluate our model on eight agent benchmarks on three different platforms, including AndroidControl-Low (Li et al., 2024), AndroidControl-High (Li et al., 2024), GUI-Odyssey (Lu et al., 2024), ScreenSpot (Cheng et al., 2024), ScreenSpot-Pro (Li et al., 2025), GUI-Act-Web (Chen et al., 2024), OmniAct-Web (Kapoor et al., 2024), and OmniAct-Desktop (Kapoor et al., 2024). We only use the test splits of these benchmarks for evaluation.

**Evaluation metrics.** Following Os-Atlas (Wu et al., 2024b), we use three commonly adopted metrics for GUI agents in evaluation: action type prediction accuracy, click point prediction accuracy, and step success rate, denoted as Type, Grounding, and SR, respectively. In more detail, Type measures the exact match score between the predicted action types (*e.g.*, 'click' and 'scroll') and the ground truth. Grounding evaluates the performance of GUI grounding in downstream tasks. Be-

Table 1: GUI grounding accuracy on ScreenSpot and ScreenSpot-Pro. All experiments are conducted under the same zero-shot prompt for fair comparison. * denotes supervised fine-tuned on GUI-R1-3K. † denotes trained with GUI-GRPO. The best results are in bold.

| Models | ScreenSpot-Pro | | | | | | | | | | | | ScreenSpot | | | |
| | Dev | | Creative | | CAD | | Scientific | | Office | | OS | | Web | | Desktop | |
| | Text | Icon | Text | Icon | Text | Icon | Text | Icon | Text | Icon | Text | Icon | Text | Icon | Text | Icon |
|---|---|---|---|---|---|---|---|---|---|---|---|---|---|---|---|---|
| Supervised Fine-Tuning | | | | | | | | | | | | | | | | |
| SeeClick | 0.6 | 0.0 | 1.0 | 0.0 | 2.5 | 0.0 | 3.5 | 0.0 | 1.1 | 0.0 | 2.8 | 0.0 | 55.7 | 32.5 | 72.2 | 30.0 |
| Os-Atlas-4B | 7.1 | 0.0 | 3.0 | 1.4 | 2.0 | 0.0 | 9.0 | 5.5 | 5.1 | 3.8 | 5.6 | 0.0 | 82.6 | 63.1 | 72.1 | 45.7 |
| ShowUI-2B | 16.9 | 1.4 | 9.1 | 0.0 | 2.5 | 0.0 | 13.2 | 7.3 | 15.3 | 7.5 | 10.3 | 2.2 | - | - | - | - |
| CogAgent-18B | 14.9 | 0.7 | 9.6 | 0.0 | 7.1 | 3.1 | 22.2 | 1.8 | 13.0 | 0.0 | 5.6 | 0.0 | 70.4 | 28.6 | 74.2 | 20.0 |
| Aria-GUI | 16.2 | 0.0 | 23.7 | 2.1 | 7.6 | 1.6 | 27.1 | 6.4 | 20.3 | 1.9 | 4.7 | 0.0 | - | - | - | - |
| UGround-7B | 26.6 | 2.1 | 27.3 | 2.8 | 14.2 | 1.6 | 31.9 | 2.7 | 31.6 | 11.3 | 17.8 | 0.0 | 80.4 | 70.4 | 82.5 | 63.6 |
| Claude** | 22.0 | 3.9 | 25.9 | 3.4 | 14.5 | 3.7 | 33.9 | 15.8 | 30.1 | 16.3 | 11.0 | 4.5 | - | - | - | - |
| Os-Atlas-7B | 33.1 | 1.4 | 28.8 | 2.8 | 12.2 | 4.7 | 37.5 | 7.3 | 33.9 | 5.7 | 27.1 | 4.5 | 90.8 | 74.2 | 91.7 | 62.8 |
| QwenVL2.5-3B* | 20.3 | 1.8 | 24.6 | 2.8 | 11.2 | 4.7 | 39.5 | 6.4 | 28.6 | 5.7 | 17.8 | 2.2 | 73.0 | 48.5 | 85.7 | 46.2 |
| QwenVL2.5-7B* | 31.4 | 1.8 | 27.3 | 3.5 | 15.7 | 5.1 | 40.7 | 7.9 | 39.7 | 8.9 | 32.4 | 6.9 | 87.8 | 68.2 | 90.3 | 62.8 |
| Zero Shot | | | | | | | | | | | | | | | | |
| QwenVL-7B | 0.0 | 0.0 | 0.0 | 0.0 | 0.0 | 0.0 | 0.7 | 0.0 | 0.0 | 0.0 | 0.0 | 0.0 | - | - | - | - |
| GPT-4o | 1.3 | 0.0 | 1.0 | 0.0 | 2.0 | 0.0 | 2.1 | 0.0 | 1.1 | 0.0 | 0.0 | 0.0 | - | - | - | - |
| QwenVL2.5-3B | 16.2 | 1.4 | 23.3 | 1.4 | 10.2 | 4.7 | 38.2 | 6.4 | 24.3 | 3.8 | 15.0 | 1.1 | 60.8 | 43.5 | 70.1 | 35.0 |
| QwenVL2.5-7B | 33.1 | 2.1 | 23.7 | 3.5 | 12.2 | 6.3 | 36.8 | 7.3 | 37.8 | 7.5 | 30.8 | 6.9 | 86.9 | 65.1 | 89.7 | 60.0 |
| Reinforcement Fine-Tuning | | | | | | | | | | | | | | | | |
| UI-R1-3B | 22.7 | 4.1 | 27.3 | 3.5 | 11.2 | 6.3 | 43.4 | 11.8 | 32.2 | 11.3 | 13.1 | 4.5 | 85.2 | 73.3 | 90.2 | 59.3 |
| GUI-R1-3B | 33.8 | 4.8 | 40.9 | 5.6 | 26.4 | 7.8 | **61.8** | **17.3** | 53.6 | 17.0 | 28.1 | 5.6 | 89.6 | 72.1 | **93.8** | 64.8 |
| GUI-R1-7B | 49.4 | 4.8 | 38.9 | 8.4 | 23.9 | 6.3 | 55.6 | 11.8 | 58.7 | 26.4 | 42.1 | 16.9 | 91.3 | 75.7 | 91.8 | 73.6 |
| GUI-R1-7B† | **52.8** | **7.6** | **44.9** | **9.1** | **33.2** | **12.5** | 58.3 | 16.4 | **65.1** | **32.5** | **46.9** | **21.3** | **92.7** | **79.8** | 93.6 | **76.4** |

sides, SR represents the step-wise success rate, where a step is deemed successful only if both the predicted action and its associated arguments (*e.g.*, point for click actions and input text for scroll actions) are correct.

## 3.3 EXPERIMENTAL RESULTS

We here evaluate our GUI-R1 model by comparing it with current state-of-the-art (SOTA) models on various tasks, including GUI grounding tasks, GUI low-level tasks, and GUI high-level tasks.

**Grounding capability.** We evaluate the grounding capability of GUI-R1 using ScreenSpot (Cheng et al., 2024) and ScreenSpot-Pro (Li et al., 2025). ScreenSpot assesses GUI grounding performance across mobile, desktop, and web platforms, while ScreenSpot-Pro focuses on high-resolution professional environments, featuring expert-annotated tasks spanning 23 applications, five industries, and three operating systems.

As shown in Table 1, compared to the previous SOTA model Os-Atlas-7B, which was trained with large-scale data using supervised fine-tuning (SFT), the RFT approach achieves superior performance on the 3B-sized Qwen2.5-VL model using only 0.2% of the data (3K vs. 14M). Furthermore, compared to the base models QwenVL2.5-3B/7B and the SFT-trained QwenVL2.5* 3B/7B models using the same dataset, the RFT-based GUI-R1 demonstrates significantly better performance in GUI grounding tasks. Moreover, at the 3B scale, GUI-R1 achieves substantial gains over SFT models on ScreenSpot (80.08 vs. 63.55) and ScreenSpot-Pro (25.23 vs. 13.80), representing improvements of 26.3% and 82.8%, respectively. This highlights the effectiveness of the RL training framework in leveraging small-scale datasets to achieve significant performance improvements, which demonstrates its potential as a data-efficient and scalable approach for model training in resource-constrained environments. Furthermore, the use of the GUI-GRPO algorithm can further enhance GUI-R1's capabilities in grounding tasks, validating our insights regarding the adaptability of reinforcement learning for GUI grounding tasks.

**Low-level task capability.** We evaluate the low-level task execution capability of GUI-R1 using four benchmark datasets: AndroidControl-Low (Li et al., 2024), GUI-Act-Web (Li et al., 2025), OmniAct-Web, and OmniAct-Desktop (Kapoor et al., 2024). AndroidControl-Low evaluates low-level task execution on mobile platforms, while GUI-Act-Web and OmniAct-Web focus on low-

Table 2: GUI low-level task accuracy on GUI-Act-Web, OmniAct-Web, OmniAct-Desktop, and AndroidControl-Low. All experiments are conducted under the same zero-shot prompt for fair comparison. * denotes supervised fine-tuned on GUI-R1-3K. The best results are in bold.

| Models | GUI-Act-Web | | | OmniAct-Web | | | OmniAct-Desktop | | | AndroidControl-Low | | | Overall |
|---|---|---|---|---|---|---|---|---|---|---|---|---|---|
| | Type | GR | SR | Type | GR | SR | Type | GR | SR | Type | GR | SR | |
| Supervised Fine-Tuning | | | | | | | | | | | | | |
| Os-Atlas-4B | 79.22 | 58.57 | 42.62 | 46.74 | 49.24 | 22.99 | 63.30 | 42.55 | 26.94 | 64.58 | 71.19 | 40.62 | 50.71 |
| Os-Atlas-7B | 86.95 | 75.61 | 57.02 | 85.63 | 69.35 | 59.15 | 90.24 | 62.87 | 56.73 | 73.00 | 73.37 | 50.94 | 70.07 |
| QwenVL2.5-3B* | 76.95 | 66.34 | 61.69 | 66.24 | 56.91 | 53.02 | 77.62 | 62.54 | 63.76 | 71.08 | 74.53 | 58.79 | 65.79 |
| QwenVL2.5-7B* | 87.66 | 84.77 | 79.89 | 81.62 | 73.45 | 73.39 | 86.23 | 80.17 | 79.80 | 84.00 | 85.74 | 64.32 | 80.09 |
| Zero Shot | | | | | | | | | | | | | |
| GPT-4o | 77.09 | 45.02 | 41.84 | 79.33 | 42.79 | 34.06 | 79.97 | 63.25 | 50.67 | 74.33 | 38.67 | 28.39 | 54.46 |
| QwenVL2.5-3B | 56.10 | 64.28 | 55.61 | 50.63 | 46.89 | 47.02 | 56.95 | 47.97 | 46.89 | 62.03 | 74.07 | 59.32 | 55.65 |
| QwenVL2.5-7B | 86.59 | 84.39 | 78.63 | 79.15 | 71.32 | 71.21 | 84.74 | 79.89 | 79.66 | 83.44 | **87.08** | 62.50 | 79.05 |
| Reinforcement Fine-Tuning | | | | | | | | | | | | | |
| UI-R1-3B | 75.89 | 79.43 | 67.31 | 75.42 | 61.35 | 61.33 | 73.41 | 64.12 | 63.98 | 79.15 | 82.41 | 66.44 | 70.85 |
| GUI-R1-3B | 89.86 | 87.42 | 76.31 | 88.58 | 75.10 | 75.08 | 91.86 | 78.37 | 78.31 | 83.68 | 81.59 | 64.41 | 80.88 |
| GUI-R1-7B | **90.85** | **88.06** | **80.31** | **91.16** | **77.29** | **77.35** | **92.20** | **83.36** | **83.33** | **85.17** | 84.02 | **66.52** | **83.30** |

level task execution on web platforms. OmniAct-Desktop is used to test low-level task execution on desktop platforms.

As demonstrated in Table 2, our method impressively improves the success rate of GUI low-level tasks for 3B and 7B models, with the average success rate increasing from 55.65 to 80.88 at the 3B scale. Compared to UI-R1 (Lu et al., 2025a), which is concurrent work also trained using RFT, our model achieves a 10-point improvement at the 3B scale, validating that RL training focused on high-level tasks can further enhance the model's understanding of low-level instructions. Note that an interesting observation is that the use of small-scale SFT data even leads to performance degradation on some metrics, such as GR on AndroidControl-Low. This limitation stems from SFT's reliance on task-specific labeled data, which constrains the model's ability to adapt to unseen environments. In contrast, our RFT method not only enhances out-of-distribution (OOD) generalization by optimizing task-specific rewards but also achieves this with fewer training examples, which provides a scalable and efficient alternative to traditional SFT methods.

**High-level task capability.** We evaluate the high-level task execution capability of GUI-R1 using AndroidControl-High (Li et al., 2024) and GUI-Odyssey (Lu et al., 2024). AndroidControl-High evaluates high-level task execution on mobile platforms, while GUI-Odyssey focuses on cross-app navigation scenarios, featuring high-level tasks spanning six applications and 203 apps.

As shown in Table 3, due to our unified action space with rule-based reward modeling, GUI-R1 achieves SOTA on high-level GUI tasks. Compared to the closed-source model GPT-4o, our 3B-scale method achieves an absolute improvement of 21.06, demonstrating that RFT, in contrast to SFT, can efficiently and reliably enhance the success rate of GUI agents in real-world tasks. Furthermore, compared to UI-R1 (Lu et al., 2025a), which focuses on improving low-level grounding capabilities, our model achieves an average improvement of 3.4 points at the 3B scale, with a particularly notable 27.2% lead in the step

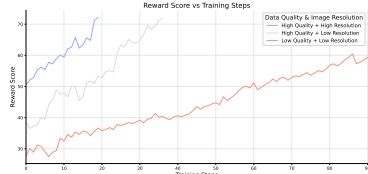

Figure 4: Ablation study of image resolution and data quality.

success rate on GUI-Odyssey. This indicates that RL training focused on low-level tasks is insufficient for handling complex high-level instructions. RFT, designed for high-level tasks, is better suited as a direction for developing GUI agent models.

Besides, in Appendix B, we explore the scalability of GUI-R1 by expanding the training data from 3K to 18K. The results show that our GUI-R1 can outperform InfiGUI-R1 (Liu et al., 2025b) with less training data (18K vs. 32K). Besides, we offer more experimental results about online agent capability evaluations, to justify the superiority of our GUI-R1.

### 3.4 ABLATION STUDY

**Image resolution and data quality.** To investigate the impact of image resolution and data quality on GUI RFT, we conduct corresponding ablation experiments, with the results shown in Figure 4.

Table 3: GUI high-level task accuracy on AndroidControl-High and GUI-Odyssey. All experiments are conducted under the same zero-shot prompt for fair comparison. * denotes supervised fine-tuned on GUI-R1-3K. The best results are in bold.

| Models | AndroidControl-High | | | GUI-Odyssey | | | Overall |
|--------|------|------|------|------|------|------|---------|
| | Type | GR | SR | Type | GR | SR | |
| Supervised Fine-Tuning | | | | | | | |
| OS-Atlas-4B | 49.01 | 49.51 | 22.77 | 49.63 | 34.63 | 20.25 | 37.63 |
| OS-Atlas-7B | 57.44 | 54.90 | 29.83 | 60.42 | 39.74 | 26.96 | 44.88 |
| QwenVL2.5-3B* | 52.05 | 49.53 | 41.22 | 43.69 | 32.21 | 27.31 | 41.00 |
| QwenVL2.5-7B* | 69.15 | 58.69 | 48.11 | 56.78 | 38.65 | 34.44 | 50.97 |
| Zero Shot | | | | | | | |
| GPT-4o | 63.06 | 30.90 | 21.17 | 37.50 | 14.17 | 5.36 | 28.69 |
| QwenVL2.5-3B | 47.81 | 46.51 | 38.90 | 37.40 | 26.49 | 26.69 | 37.30 |
| QwenVL2.5-7B | 68.67 | 59.71 | 47.06 | 55.60 | 37.78 | 34.37 | 50.53 |
| Reinforcement Fine-Tuning | | | | | | | |
| UI-R1-3B | 57.85 | 55.70 | 45.44 | 52.16 | 34.46 | 32.49 | 46.35 |
| GUI-R1-3B | 58.04 | 56.24 | 46.55 | 54.84 | 41.52 | **41.33** | 49.75 |
| GUI-R1-7B | **71.63** | **65.56** | **51.67** | **65.49** | **43.64** | 38.79 | **56.13** |

As observed, when using the filtered GUI-R1-3K dataset, the model requires only a few updates to achieve relatively high rewards. In contrast, training with unfiltered and low-quality data necessitates significantly more training time for the model to converge, with a noticeably lower performance ceiling. To further explore the effect of image resolution on model training, we increase the image resolution to twice its original size (from 1,048,576 pixels to 2,097,152 pixels). As shown in Figure 4, because of the high resolution of GUI task images and the small size of many UI elements, increasing the image resolution allows the model to perceive these elements more clearly, which accelerates the convergence speed of RFT and improves the performance ceiling.

**Coefficients in the reward function.** To explore the impact of the coefficients for format rewards and accuracy rewards in the reward function on the final performance, we conduct relevant ablation experiments, as shown in Table 4. The results indicate that reducing the coefficient ratio of format rewards

Table 4: Ablation study of the coefficient $\alpha$ and $\beta$ in the reward function. The best results are in bold.

| $\alpha$ | $\beta$ | AndroidControl-High | | | GUI-Odyssey | | | Overall |
|----------|---------|------|------|------|------|------|------|---------|
| | | Type | GR | SR | Type | GR | SR | |
| 0.2 | 0.8 | **58.04** | **56.24** | 46.55 | **54.84** | **41.52** | **41.33** | **49.75** |
| 0.5 | 0.5 | 57.93 | 55.91 | **46.62** | 52.77 | 37.44 | 35.66 | 47.72 |
| 0.8 | 0.2 | 57.85 | 55.70 | 45.44 | 52.16 | 34.46 | 32.49 | 46.48 |

leads to consistent performance improvements. This is because format rewards are easier to learn during training and often converge early in the process. By amplifying the accuracy rewards, the advantages of providing correct answers are further emphasized, ultimately leading to more performance improvements.

Note that we also include the ablation studies about the reward scaling strategy and multi-image inputs. Overall, these studies demonstrate that our model can benefit from learning from those difficult examples. Besides, a single-image input for training is a more feasible and efficient strategy than multi-image inputs. Detailed settings, results, and analysis can be checked in Appendix C.

## 4 CONCLUSION

This paper presents GUI-R1, which is the first GUI reinforcement learning framework grounded in unified action space rule modeling. By integrating reinforcement fine-tuning with large vision-language models, GUI-R1 enables effective contextual action prediction and verifiable reward-driven learning in GUI environments. Extensive experiments demonstrate that GUI-R1 consistently outperforms baselines on various tasks. It exhibits strong generalization capabilities, making it well-suited for real-world applications where interaction complexity and dynamic changes are prevalent. Moving forward, we plan to extend GUI-R1 to support collaborative multi-agent interaction and robust error correction policies, enabling the system to handle complex tasks with greater scalability.

ETHICS STATEMENT

GUI-R1 for enhancing GUI grounding capabilities is designed with careful consideration of ethical implications. All GUI interaction data used for training is collected from publicly available interfaces or synthetic environments, ensuring no violation of user privacy or proprietary software terms. We do not collect, store, or utilize any personal user data, sensitive information, or confidential interface content during the training process.

REPRODUCIBILITY STATEMENT

Experimental settings are carefully described and listed in Section 3. To further ensure reproducibility, we promise to release training details, data protocols, and open-source both the code and model checkpoints.

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

# Appendix

Table 5: GUI grounding accuracy on ScreenSpot-Pro. When scaling the data from 3K to 18K, GUI-R1 achieves more competitive performance. The best results are in bold.

| Models | Dev | | Creative | | CAD | | Scientific | | Office | | OS | | Overall |
|---|---|---|---|---|---|---|---|---|---|---|---|---|---|
| | Text | Icon | Text | Icon | Text | Icon | Text | Icon | Text | Icon | Text | Icon | |
| Supervised Fine-Tuning | | | | | | | | | | | | | |
| SeeClick | 0.6 | 0.0 | 1.0 | 0.0 | 2.5 | 0.0 | 3.5 | 0.0 | 1.1 | 0.0 | 2.8 | 0.0 | 1.0 |
| Os-Atlas-4B | 7.1 | 0.0 | 3.0 | 1.4 | 2.0 | 0.0 | 9.0 | 5.5 | 5.1 | 3.8 | 5.6 | 0.0 | 3.5 |
| ShowUI-2B | 16.9 | 1.4 | 9.1 | 0.0 | 2.5 | 0.0 | 13.2 | 7.3 | 15.3 | 7.5 | 10.3 | 2.2 | 7.1 |
| CogAgent-18B | 14.9 | 0.7 | 9.6 | 0.0 | 7.1 | 3.1 | 22.2 | 1.8 | 13.0 | 0.0 | 5.6 | 0.0 | 6.5 |
| Aria-GUI | 16.2 | 0.0 | 23.7 | 2.1 | 7.6 | 1.6 | 27.1 | 6.4 | 20.3 | 1.9 | 4.7 | 0.0 | 9.3 |
| UGround-7B | 26.6 | 2.1 | 27.3 | 2.8 | 14.2 | 1.6 | 31.9 | 2.7 | 31.6 | 11.3 | 17.8 | 0.0 | 14.2 |
| Os-Atlas-7B | 33.1 | 1.4 | 28.8 | 2.8 | 12.2 | 4.7 | 37.5 | 7.3 | 33.9 | 5.7 | 27.1 | 4.5 | 16.6 |
| UI-TARS-2B | 47.4 | 4.1 | 42.9 | 6.3 | 17.8 | 4.7 | 56.9 | 17.3 | 50.3 | 17.0 | 21.5 | 5.6 | 24.3 |
| UI-TARS-7B | **58.4** | **12.4** | **50.0** | **9.1** | 20.8 | 9.4 | **63.9** | **31.8** | 63.3 | 20.8 | 30.8 | **16.9** | 32.3 |
| Reinforcement Fine-Tuning | | | | | | | | | | | | | |
| UI-R1-3B | 22.7 | 4.1 | 27.3 | 3.5 | 11.2 | 6.3 | 43.4 | 11.8 | 32.2 | 11.3 | 13.1 | 4.5 | 16.0 |
| GUI-R1-3B | 33.8 | 4.8 | 40.9 | 5.6 | 26.4 | 7.8 | 61.8 | 17.3 | 53.6 | 17.0 | 28.1 | 5.6 | 25.2 |
| InfiGUI-R1-3B | 51.3 | **12.4** | 44.9 | 7.0 | **33.0** | **14.1** | 58.3 | 20.0 | 65.5 | 28.3 | 43.9 | 12.4 | 32.6 |
| GUI-R1-3B-18K | 52.7 | **12.4** | 46.5 | 8.4 | 31.4 | 12.3 | 62.1 | 26.3 | **66.3** | **29.0** | **44.6** | 15.0 | **33.9** |

# A    RELATED WORK

**GUI agents.**    Autonomous agents driven by large foundation models (*e.g.*, large language models (LLMs) and large vision-language models (LVLMs)) have gained significant attention for their powerful interactive capabilities (Sumers et al., 2023; Tang et al., 2025; Zhang et al., 2025b;a; Liu et al., 2025a; Li et al., 2023; Wu et al., 2024a; Mialon et al., 2023; Huang et al., 2024; Shao et al., 2023; Sun et al., 2024; Yao et al., 2023). These operating systems via programs or API calls (Wang et al., 2024a; Sun et al., 2023). However, the closed-source nature of most commercial software limits access to internal APIs or code, which promotes a shift in research toward GUI agents. Different from traditional programmatic agents, GUI agents simulate human interactions via mouse and keyboard inputs, which enable broader flexibility in solving complex tasks. Recent works have advanced this direction. For instance, UGround (Gou et al., 2024) developed a specialized GUI grounding model for precise GUI element localization. OS-Atlas (Wu et al., 2024b) introduced large action models to handle general agent tasks by interpreting human intentions and predicting actions in the form of function calls. UITars (Qin et al., 2025) proposed a more comprehensive method by combining GUI-related pretraining with task-level reasoning fine-tuning to better capture the complexity of GUI interactions. Nevertheless, these methods all rely on the paradigm of supervised fine-tuning (SFT), which suffers from two main limitations: (1) the training process requires vast amounts of diverse data; (2) the models exhibit limited generalization capabilities, which struggle to understand GUI screenshots and adapt to unseen interfaces. These limitations motivate the development of a more advanced learning paradigm for GUI agents beyond traditional SFT methods.

**Reinforcement fine-tuning.**    Rule-based reinforcement fine-tuning, exemplified by OpenAI o1 (Jaech et al., 2024) and DeepSeek-R1 (Guo et al., 2025), has demonstrated strong performance in mathematical reasoning (Shao et al., 2024), code generation (Liu & Zhang, 2025), and multi-step logic tasks (Wang* et al., 2025). Subsequent studies have extended this paradigm to multimodal models by designing task-specific reward functions for vision-based tasks, such as correct class prediction in image classification (Pan & Liu, 2025; Chen et al., 2025b; Meng et al., 2025), intersection-over-union (IoU) metrics in image localization and detection (Huang et al., 2025; Liu et al., 2025c), and accurate click position prediction in low-level GUI grounding tasks (Lu et al., 2025a). These works demonstrate that verifiable reward signals, *e.g.*, symbolic correctness or execution-based feedback, can effectively substitute traditional supervision. Despite the strong potential of RFT in various tasks, it remains underexplored in complex high-level GUI agent tasks. Compared to other domains, building intelligent agents for high-level GUI tasks is particularly challenging due to diverse UI layouts, implicit task semantics, and long-horizon action dependencies. This imposes higher demands on the model's contextual learning and understanding capabilities. To the best of

our knowledge, GUI-R1 is the first RFT-based framework specifically designed for high-level GUI agents. Extensive subsequent works (Liu et al., 2025b; Zhou et al., 2025; Lu et al., 2025b;a) validate the forward-looking insights of GUI-R1 in pioneering the paradigm shift in GUI training from SFT to RL.

## B  ADDITIONAL EXPERIMENTS

**Scalability evaluation** To explore the scalability of GUI-R1. Specifically, we expanded the training data from 3K to 18K. The experimental results are shown in Table 5 and Table 6. As we can see, compared to the state-of-the-art (SOTA) supervised fine-tuning method UI-TARS-7B, GUI-R1-3B-18K achieves competitive results and higher overall localization performance on both ScreenSpot and ScreenSpot-Pro. Furthermore, compared to the state-of-the-art reinforcement learning-trained method InfiGUI-R1, GUI-R1-3B-18K achieves better performance with a smaller amount of training data (18K vs. 32K), demonstrating the effectiveness and efficiency of our method. These results highlight the excellent scalability of GUI-R1.

**Online agent capability evaluation.** To explore the performance of GUI-R1 in dynamic environment tests, we evaluate it on two widely used online benchmarks: OSWorld (Xie et al., 2024) and AndroidWorld (Rawles et al., 2024). Specifically, OSWorld provides a scalable and diverse environment for assessing multimodal agents on complex tasks across Ubuntu, Windows, and macOS platforms. It consists of 369 tasks involving real-world web and desktop applications, accompanied by detailed setups and evaluation scripts. The evaluation is conducted in screenshot-only mode. To mitigate potential interference from network instability and environmental factors, the final score is averaged over three runs. Additionally, tasks where the model decides to execute "CallUser" or fails to output "Finish" are considered infeasible for evaluation. AndroidWorld is an environment designed specifically for developing and benchmarking autonomous agents on a live Android emulator. It includes 116 tasks across 20 mobile apps, with dynamic task variations generated through randomized parameters. This dataset is ideal for evaluating agents' adaptability and planning capabilities in mobile environments.

Experimental results are provided in Table 7. As can be seen, on OSWorld (Xie et al., 2024), with a budget of 15 steps, GUI-R1 outperformed UI-TARS-7B, highlighting the efficiency and effectiveness of reinforcement learning in enhancing the model's ability to tackle complex desktop-based tasks. Afterward, on AndroidWorld (Rawles et al., 2024), a similar conclusion was reached, with GUI-R1 achieving a competitive score of 34.3. Overall, these results validate the potential of reinforcement learning in enabling agent models to excel in reasoning-intensive tasks. It demonstrates the ability to efficiently address challenges in online environments with limited training data, paving the way for multi-turn interactions with realistic situations.

## C  ADDITIONAL ABLATION STUDIES

**Reward scale.** To enhance the model's performance, we introduce a reward scaling strategy. For smaller buttons and elements, the model often struggles to accurately predict their positions. Therefore, when the model successfully predicts these challenging cases, we appropriately amplify their reward weights to encourage the model to learn from difficult examples. To validate the effectiveness of the reward scaling strategy, we conduct ablation experiments. As shown in Table 8, the model's performance improves further when the reward scaling strategy is applied.

**Multi-image inputs.** We investigate the impact of multi-image inputs on GUI-R1's performance in high-level tasks. To improve efficiency, for trajectory data of high-level tasks, we only use the historical action sequence and the current screenshot as input for training. This way significantly reduces the training overhead caused by multi-image inputs. Experimental results are shown in Table 8. As we can see, when all images from the historical trajectory are used for training, the training overhead increases significantly, nearly fivefold. However, the performance improvement is only 2%, which is highly inefficient. Therefore, using single-image input for training is a more feasible and efficient strategy.

**GUI-GRPO.** To validate the effectiveness of GUI-GRPO, we conduct comprehensive ablation experiments comparing different reinforcement learning algorithms. As shown in Table 9, our pro-

Table 6: GUI grounding accuracy on ScreenSpot. When scaling the data from 3K to 18K, GUI-R1 achieves more competitive performance. The best results are in bold.

| Models | Web | | Desktop | | Mobile | | Overall |
|---|---|---|---|---|---|---|---|
| | Text | Icon | Text | Icon | Text | Icon | |
| Supervised Fine-Tuning | | | | | | | |
| SeeClick | 55.7 | 32.5 | 72.2 | 30.0 | 78.0 | 52.0 | 53.4 |
| Os-Atlas-4B | 82.6 | 63.1 | 72.1 | 45.7 | 93.0 | 72.9 | 71.6 |
| ShowUI-2B | 81.7 | 63.6 | 76.3 | 61.1 | 92.3 | 75.5 | 75.1 |
| CogAgent-18B | 70.4 | 28.6 | 74.2 | 20.0 | 67.0 | 24.0 | 47.4 |
| UGround-7B | 80.4 | 70.4 | 82.5 | 63.6 | 82.8 | 60.3 | 73.3 |
| Os-Atlas-7B | 90.8 | 74.2 | 91.7 | 62.8 | 93.0 | 72.9 | 80.9 |
| UI-TARS-2B | 84.3 | 74.8 | 90.7 | 68.6 | 93.0 | 75.5 | 81.2 |
| Reinforcement Fine-Tuning | | | | | | | |
| UI-R1-3B | 85.2 | 73.3 | 90.2 | 59.3 | - | - | 77.0 |
| GUI-R1-3B | 89.6 | 72.1 | 93.8 | 64.8 | 95.6 | 79.7 | 82.6 |
| InfiGUI-R1-3B | 91.7 | **77.6** | 94.3 | 77.1 | **97.1** | 81.2 | 86.5 |
| GUI-R1-3B-18K | **91.8** | 76.2 | **95.5** | **78.2** | 96.9 | **81.4** | **86.7** |

posed GUI-GRPO demonstrates significant improvements in GUI grounding capabilities compared to baseline methods. Specifically, GUI-GRPO substantially outperforms both standard PPO and vanilla GRPO across all evaluated GUI tasks, achieving notable performance gains that highlight the importance of our tailored modifications. The results consistently show that GUI-GRPO's entropy regularization and sentence-level optimization effectively address the unique challenges in GUI grounding scenarios, leading to more robust and accurate spatial understanding. These findings confirm that our algorithm-specific enhancements are crucial for maximizing the reinforcement learning effectiveness in GUI-related tasks.

**Reward Type.** We investigate the impact of reward type on GUI-R1's performance in high-level tasks. As shown in Table 10, changing $R_{point}$ to a continued gaussian form significantly boosts performance, while the impact of changing $R_{text}$ is smaller. This is because GUI input text is typically a word or short phrase, and an F1 score threshold of 0.5 already yields highly accurate rewards.

**F1 Score Threshold.** As shown in Table 11, it can be observed that increasing the F1 Score threshold has minimal impact on the final performance. Setting the threshold to 0.5 is sufficient to provide accurate and reliable rewards for text input matching.

Table 7: Online agent capability evaluation.

| Method | OSWorld | AndroidWorld |
|---|---|---|
| GPT-4o | 5.0 | 34.5 |
| Gemini-Pro-1.5 | 5.4 | 22.8 |
| CogAgent-9B | 8.1 | - |
| Claude Computer-Use | 14.9 (15 steps) | 27.9 |
| UGround | - | 32.8 |
| Aria-UI | 15.2 | 44.8 |
| Aguvis-7B | 14.8 | 37.1 |
| Aguvis-72B | 17.0 | - |
| OS-Atlas-7B | 14.6 | - |
| UI-TARS-7B | 17.7 (15 steps) | 33.0 |
| GUI-R1-3B-18K | 18.2 (15 steps) | 34.3 |

## D  VISUALIZATION

In Figure 5, we provide additional visualization of the training process. As shown in Figure 5a and Figure 5b, it can be observed that the format reward converges quickly in the early stages of training,

Table 8: Ablation study of the reward scale and multi-image inputs. The best results are in bold.

| Multi-Image Inputs | Reward Scale | GPU Hours | AndroidControl-High | | | GUI-Odyssey | | | Overall |
| --- | --- | --- | --- | --- | --- | --- | --- | --- | --- |
| | | | Type | GR | SR | Type | GR | SR | |
| ✗ | ✗ | 10×8 | 58.04 | 56.24 | 46.55 | 54.84 | 41.52 | 41.33 | 49.75 |
| ✗ | ✓ | 10×8 | 58.32 | 56.45 | **46.78** | 55.02 | 41.76 | 41.89 | 50.03 |
| ✓ | ✓ | 52×8 | **59.32** | **56.98** | 46.62 | **55.67** | **42.48** | **43.66** | **50.79** |

Table 9: Ablation study of RL methods, including GUI-GRPO (Ours), GRPO, and PPO. The best results are in bold.

| RL Method | AndroidControl-High | | | GUI-Odyssey | | |
| --- | --- | --- | --- | --- | --- | --- |
| | Type | GR | SR | Type | GR | SR |
| GUI-GRPO | **58.04** | **56.24** | **46.55** | **54.84** | **41.52** | **41.33** |
| GRPO | 55.43 | 54.76 | 41.23 | 51.12 | 37.97 | 37.26 |
| PPO | 49.57 | 47.33 | 39.98 | 44.45 | 32.81 | 32.73 |

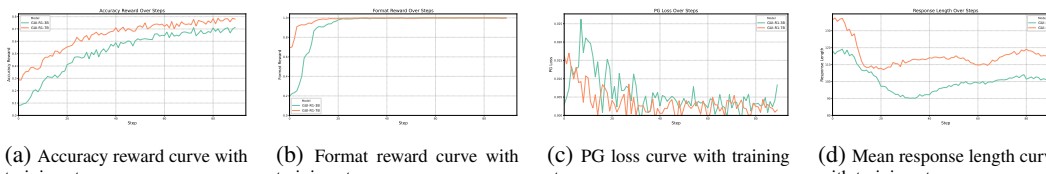

(a) Accuracy reward curve with training steps.

(b) Format reward curve with training steps.

(c) PG loss curve with training steps.

(d) Mean response length curve with training steps.

Figure 5: Visualization of the training process of GUI-R1. To provide more details, we report the curves of GUI-R1's key metrics during training, including format reward, accuracy reward, mean response length, and policy gradient (PG) loss, as they vary with the training steps.

while the accuracy reward becomes the main source of differentiated rewards in the later stages of training. Furthermore, as illustrated in Figure 5d, the mean response length first decreases and then gradually increases, but the "aha moment" does not occur. This may be due to the single-image input training method in a non-interactive environment, which prevents the model from autonomously tracing back the sequence of incorrect actions. Exploring multi-image high-level tasks in interactive environments could be a potential direction for inducing the emergence of the "aha moment" in future research. We also provide some specific test cases for visualization (see Appendix E), which further illustrate how our GUI-R1 effectively completes the tasks with high precision and reliability, showcasing its practical deployment capabilities.

## E   SPECIFIC CASES

We list some cases executed by GUI-R1 in Figure 6 and Figure 7 to help readers better understand the superiority of our GUI-R1. GUI-R1 can autonomously interact with the GUI and environment in a fully automated manner, similar to humans. It is capable of completing complex high-level tasks on behalf of humans, thereby improving efficiency.

## F   BROADER IMPACTS

GUI agents represent a transformative advancement in human-computer interaction by enabling intuitive, language-driven control of software applications. Through automation of repetitive tasks, streamlined software navigation, and cross-platform adaptability, GUI Agents enhance user productivity and accessibility, particularly for individuals with disabilities or limited technical expertise. Moreover, their ability to autonomously explore and test interfaces accelerates software development and ensures greater reliability. However, the use of GUI agents also introduces ethical considerations regarding data privacy and potential workforce displacement in roles traditionally dependent on manual interface operations. Addressing these concerns responsibly is crucial to maximizing the societal benefits of GUI agent technologies.

Table 10: Ablation study of the reward function. The best results are in bold.

| $R_{\text{point}}$ | $R_{\text{text}}$ | AndroidControl-High | | | GUI-Odyssey | | |
|---|---|---|---|---|---|---|---|
| | | Type | GR | SR | Type | GR | SR |
| Norm | Norm | 58.04 | 56.24 | 46.55 | 54.84 | 41.52 | 41.33 |
| Norm | Gaussian | 58.13 | 56.27 | 46.63 | 54.73 | 41.50 | 41.37 |
| Gaussian | Gaussian | **58.68** | **56.72** | **47.21** | **55.22** | **42.21** | **42.14** |

Table 11: Ablation study of the F1 Score Threshold.

| Threshold | AndroidControl-High | | | GUI-Odyssey | | |
|---|---|---|---|---|---|---|
| | Type | GR | SR | Type | GR | SR |
| 0.5 | 58.04 | 56.24 | 46.55 | 54.84 | 41.52 | 41.33 |
| 0.6 | 58.09 | 56.31 | 46.47 | 54.91 | 41.61 | 41.26 |
| 0.7 | 58.13 | 56.27 | 46.63 | 54.73 | 41.50 | 41.37 |
| 0.8 | 58.01 | 56.30 | 46.52 | 54.79 | 41.48 | 41.42 |

# G    THE USE OF LARGE LANGUAGE MODELS

We declare that large language models (LLMs) were employed to assist with the refinement of this manuscript, specifically, for grammar checking, language polishing, and improving the clarity and fluency of the text. Additionally, LLMs were used in a limited capacity for minor debugging and syntactic correction of code snippets included in the work.

Instruction: Share my favorite Book "the Queen\'s Gambit" to my Friend Natalie larson over her gmail address -natalie.larson1998@gmail.com from the PocketBook app.

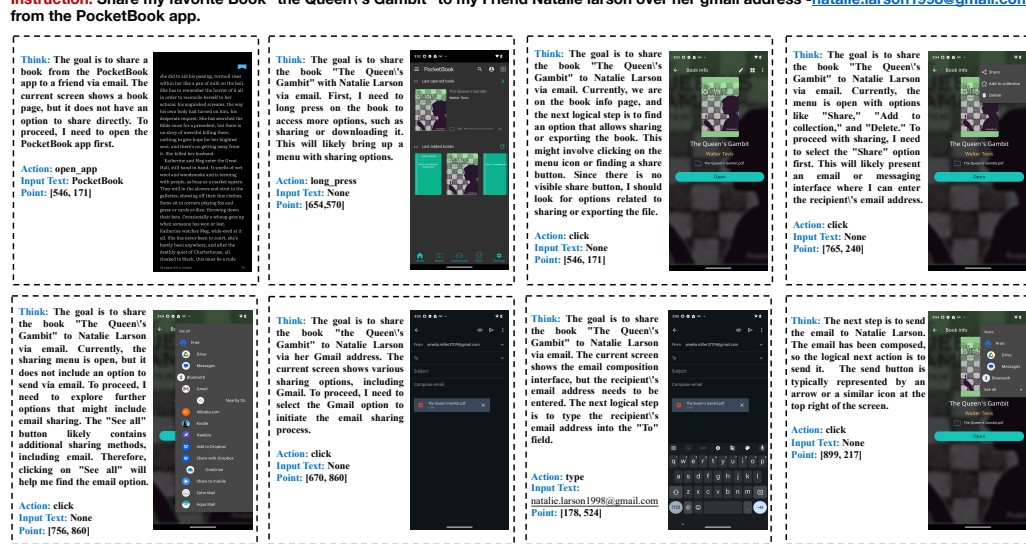

Figure 6: **Test case on Android from GUI-R1.**
The task is: `Share my favorite Book "the Queenś Gambit" to my Friend Natalie larson over her gmail address -natalie.larson1998@gmail.com from the PocketBook app.`

**Instruction:** Open the VSCode website and download the Linux version of VSCode.

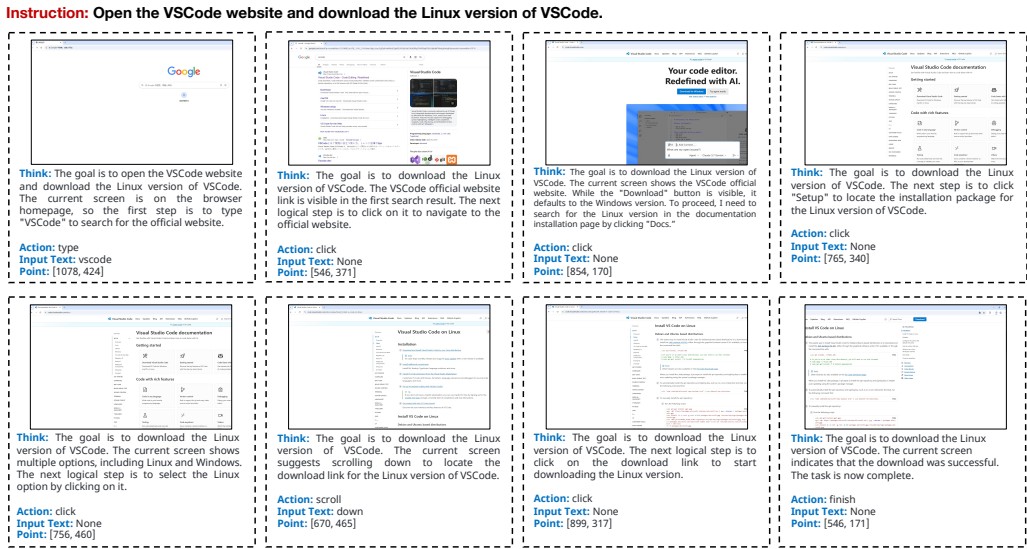

Figure 7: **Test case on Windows from GUI-R1.** The task is: `Open the VSCode website and download the Linux version of VSCode.`