# OpenReview forum: "GUI-R1: A Generalist R1-Style Vision-Language Action Model For GUI Agents"
_ICLR.cc/2026/Conference — Submitted to ICLR 2026_

### Official Review · Reviewer_rVpK · 2025-10-22

**Soundness:** 3
**Presentation:** 2
**Contribution:** 2
**Rating:** 4
**Confidence:** 4

**Summary:**

This paper proposes the first RL framework designed to enhance the GUI agents' high-level capabilities. The first contribution lies in the high-quality dataset curation method. GUI-R1-3B and GUI-R1-7B achieve great performance among various benchmarks (grounding, low-level, high-level and online). The second contribution lies in GUI-GRPO, which surpasses GRPO and PPO on GUI agent training. GUI-GRPO adds maximum entropy constraint terms to encourage exploration and changes the training granularity from token-level to sentence-level.

**Strengths:**

1. The writing is clear and easy to follow.
2. The chosen benchmarks are comprehensive and sufficient for evaluation.
3. Ablation studies are thorough and provide solid support for the claims.
4. GUI-GRPO effectively addresses key challenges in RL training for GUI agents.

**Weaknesses:**

1. **Novelty**: My primary concerns pertain to the reward design and the claimed innovations.
   - **Reward Functions**: The reward function design for GUI tasks in this work appears to lack substantial novelty. Recent advances such as VLM-R1 [1] and Visual-RFT [2] have already extended DeepSeek-R1-style reward mechanisms to multi-modal scenarios. Works like UI-R1 [3] and InfiGUI-R1 [4] introduce similar reward functions to improve action prediction accuracy, which still heavily depend on labeled data. Furthermore, SE-GUI [5], GUI-G1 [6], and GUI-G$^2$ [7] have explored non-binary and Gaussian-based rewards to strengthen GUI grounding capabilities.
   - **GUI-GRPO**: The proposed GUI-GRPO incorporates maximum entropy constraint terms to encourage exploration; however, this concept was initially explored in CISPO [8]. For sequence-level optimization, approaches such as GSPO [9] and GRPO-S [10] have proposed this prior to this work. As a result, the claimed novelty of GUI-GRPO is limited and the improvement is not specific to the GUI domain. The paper also lacks appropriate citations and further discussion of these related works.
   - **High-Level Tasks**: The only difference between high-level and low-level tasks is the addition of historical ground truth actions as prompts. The interactions remain static and single-turn, and significant limitations regarding multi-turn scenarios are not resolved.

2. **Lack of Baselines**: The experimental evaluation would be more convincing with comparisons to additional open-source models, such as InfiGUI-R1 [4] and UI-Tars [11].

**References:**

[1] VLM-R1: A Stable and Generalizable R1-style Large Vision-Language Model
[2] Visual-RFT: Visual Reinforcement Fine-Tuning
[3] UI-R1: Enhancing Efficient Action Prediction of GUI Agents by Reinforcement Learning
[4] InfiGUI-R1: Advancing Multimodal GUI Agents from Reactive Actors to Deliberative Reasoners
[5] Enhancing Visual Grounding for GUI Agents via Self-Evolutionary Reinforcement Learning
[6] GUI-G1: Understanding R1-Zero-Like Training for Visual Grounding in GUI Agents
[7] GUI-G2: Gaussian Reward Modeling for GUI Grounding
[8] MiniMax-M1: Scaling Test-Time Compute Efficiently with Lightning Attention
[9] Group Sequence Policy Optimization
[10] GTPO and GRPO-S: Token and Sequence-Level Reward Shaping with Policy Entropy
[11] UI-TARS: Pioneering Automated GUI Interaction with Native Agents

**Questions:**

See concerns mentioned above, and I have the following questions:

1. Could you provide more comprehensive ablation studies regarding the two strategies proposed in GUI-GRPO?
2. Could you include comparisons of training curves and entropy curves among PPO, GRPO, and GUI-GRPO to better illustrate the effectiveness of GUI-GRPO?

---

### Official Review · Reviewer_HMF9 · 2025-10-31

**Soundness:** 3
**Presentation:** 2
**Contribution:** 3
**Rating:** 4
**Confidence:** 4

**Summary:**

This paper proposes GUI-R1, a rule-based reinforcement learning framework for training generalist GUI agents. They also introduce a rule-based unified action space and a crafted high-quality dataset.

**Strengths:**

- One of the significant strength is the demonstration of extreme data efficiency. By curating a 3K dataset (GUI-R1-3K) and using RFT, the authors achieve SOTA performance, surpassing models trained with orders of magnitude more data (e.g., 3K vs. 13M for OS-Atlas). This illuminates the important of crafting high quality trajectory data in scaling GUI agents.
- The paper thoughtfully identifies unique challenges of the GUI domain and proposes a unified action space and verifiable rewards with a novel algorithm, GUI-GRPO, to solve them.
- The data filtering strategy is an original contribution. By using a pre-trained model to generate responses and then filtering based on the reward function (keeping samples where $0 < \text{accuracy} < 1$), the authors create a dataset with a high concentration of "learnable" examples. The ablation in Figure 4 (comparing filtered vs. unfiltered data) confirms this strategy is critical for rapid convergence and a higher performance ceiling.
- The paper's evaluation is comprehensive and convincing. It spans 8 benchmarks, 3 platforms, and 3 task granularities. The inclusion of SFT-on-3K-data as a baseline is a critical control that isolates the benefit of RFT. The multitude of ablation studies (data quality, image resolution, reward components, algorithm choice) demonstrate a deep understanding of the method.

**Weaknesses:**

- Typo at L293-294: “OS-Otlas” should be “OS-Atlas”
- The rationale of the reward design in 2.2 is not well justified. Different components of the reward are isolated. Lack of a unified equation and aggregation of the sub components.
- In 3.1 you briefly mention the data collection and data filtering process. But the details of the rationale behind the claimed data preparation method are omitted.
- Insufficient ablation study. Lack of ablation study on two components (Adding maximum entropy & sentence-level training granularity) introduced in the GUI-GRPO.

**Questions:**

- In 3.1, you mention that “To filter out low-quality data for efficient RFT, we use the Qwen2.5VL-7B (Bai et al., 2025) model to generate 10 responses for each example”. Could you clarify that how many and what kinds of examples do you use to generate responses from Qwen2.5VL-7B? Can you share more details of this process? Why the proportion of the high-level and low-level examples after data filtering seem to be so imbalance?
- Your results in Table 9 show that GUI-GRPO outperforms standard GRPO. However, this ablation combines two distinct modifications: the maximum entropy constraint and sentence-level training granularity. Could you provide an ablation that disentangles these two components? Specifically, what is the performance of:
    - (a) GRPO + maximum entropy constraint (token-level)?
    - (b) GRPO + sentence-level granularity (no max entropy)? This would clarify the individual contribution of each component.

---

### Official Review · Reviewer_z3nN · 2025-10-31

**Soundness:** 3
**Presentation:** 3
**Contribution:** 2
**Rating:** 6
**Confidence:** 2

**Summary:**

The paper introduces GUI-R1, a vision-language model trained with reinforcement learning to perform complex tasks on graphical user interfaces across platforms like Windows, Linux, Android, and the web. Unlike previous models that require millions of labeled examples, GUI-R1 learns effectively from just 3,000 high-quality examples using a rule-based reward system. It achieves state-of-the-art results on eight benchmarks, outperforming larger models while using far less data.

**Strengths:**

- GUI-R1 outperforms larger models like OS-Atlas and GPT-4o using only 3,000 training examples. The performance improvement is from using RL with well-designed rewards.
- Results on 8 benchmarks show clear improvements across grounding, low-level, and high-level tasks. The model also performs well across various platform, even on unseen interfaces and tasks.

**Weaknesses:**

- The paper claims that unifying the action space helps cross-platform generalization, but it doesn’t test this directly. What would the action space look like if we do not unify? It would be helpful if we can compare to a non-unified baseline. In addition, unified action space has been proposed in ShowUI [1].

    [1] Lin, Kevin Qinghong, et al. "Showui: One vision-language-action model for gui visual agent." Proceedings of the Computer Vision and Pattern Recognition Conference. 2025.
- RFT usually is sensitive to how diverse or confident the trajectory pool is, however trajectory sampling is usually pretty example for GUI tasks. it would be helpful to understand how the number of trajectories sampled during training (e.g., 1 vs 5 vs 10) affects performance.
- There’s no ablation on whether the <think>…</think> reasoning trace helps, hurts, or stays the same. Since the model is rewarded on both reasoning format and final action, it’s useful to test if including reasoning actually improves learning or can be dropped. Especially in the visualization examples, the reasoning traces are usually reiterating the goal, and describing what to do next, feels a little redundant given action is then being predicted.

**Questions:**

Besides the concerns I raised in weakness section above, below are other minor questions I have:

- How critical is the quality of the small training set? The paper shows that 3K curated examples are enough—but how important is that curation step? The paper shows results for 3K vs 18K training examples, but it doesn’t test how sensitive the model is to data quality. For example, what happens if we skip their reward-based filtering and use randomly sampled or noisy data? This would help justify the claim that careful curation matters.

- Among the compared baselines, only UI-R1 has been trained with RL. It would be helpful if the authors can add deeper analysis in the key differences between GUI-R1 and UI-R1.

---

### Official Review · Reviewer_Ney8 · 2025-11-02

**Soundness:** 2
**Presentation:** 3
**Contribution:** 2
**Rating:** 2
**Confidence:** 5

**Summary:**

This paper proposes to incorporate RL algorithm to enhance the LLM capabilities in GUI domains, named GUI-R1. It only requires a small amount of high-quality data across multiple domains and achieves the comparable performances with previous SOTA models.

**Strengths:**

1. It is the very early attempt to apply R1-style algorithm in GUI Agent domains.
2. It leverages the rule-based unified action space reward function to construct GUI-R1-3K datasets, which is a good contribution to the community.
3. The experiments cover multiple domains.

**Weaknesses:**

1. The paper overclaims GUI-R1 as "the first reinforcement learning framework designed to enhance the GUI capabilities of LVLMs". Some early works have attempted to apply RL to GUI tasks (e.g., Mobile platform). GUI-R1 is only the early attempt to apply R1-style (rule-based reward) RL to GUI tasks.

2. The experiments lack some important baselines, such as UI-Tars series, JEDI, InfGUL-R1, GUI-G1, GUI-Owl. The comparisons with OS-Atlas and UGround are unfair, because these early works use weak VLMs as the base., for example, OS-Atlas is based on Qwen2-VL.

3. The overall technical contributions are limited. Simply applying GRPO-style loss function in GUI is not difficult. And this kind of single-turn RL has an obvious upper boundary in real applications. Actually, this paper does not evaluate on more challenging tasks / online environments.

**Questions:**

1. Could you provide the average performance in Table 1, which can make it clear for comparison ?
2. How does the trained model perform on online benchmarks (e.g., Android World, OS-World) ?
3. Have you tried on further scaling the data or scaling the model size ? How is the performance ?

---

### Meta-Review · Area_Chair_3jp4 · 2025-12-30

**Summary:**

This submission proposes GUI-R1, a rule-based RFT approach for training GUI agents using a unified action space and verifiable reward signals, together with a small curated dataset (3K) and a customized GRPO variant GUI-GRPO. Reviewers broadly agree the direction is timely and that the empirical results suggest strong data efficiency and broad evaluation coverage, but they raise multiple concerns that affect the recommendation.

The most influential concerns are: (i) overstated novelty (“first RL framework…”) and insufficient differentiation from closely related concurrent/prior work; (ii) baseline coverage and fairness (missing/uneven comparisons, including concerns about differing base VLM backbones); and (iii) missing ablations to support key causal claims (benefit of unified action space, benefit of reasoning traces, sensitivity to trajectory sampling, and disentangling the two proposed GUI-GRPO modifications). These issues are emphasized especially strongly by the rejecting reviewer and reiterated by the two marginal-reject reviewers.

**Reviewer Concerns:**

No rebuttal is submitted.

**Reviewer Scores:**

No rebuttal is submitted. Reviewers will likely keep the initial negative rating.

Ney8: 2 → 2 (unchanged)
z3nN: 6 → 6 (unchanged)
HMF9: 4 → 4 (unchanged)
rVpK: 4 → 4 (unchanged)

---

### Decision · Program_Chairs · 2026-01-26

Reject